# Surgical intervention for paediatric infusion-related extravasation injury: a systematic review

Max Little ⓘ ,[1] Sophie Dupré,[2] Justin Conrad Rosen Wormald ⓘ ,[2,3] Matthew Gardiner,[2,4] Chris Gale ⓘ ,[5] Abhilash Jain[2,6]

[1]Trauma & Orthopaedic Surgery, Whittington Hospital NHS Trust, London, UK
[2]Plastic and Reconstructive Surgery, Nuffield Department of Orthopaedics Rheumatology and Musculoskeletal Sciences, University of Oxford, Oxford, UK
[3]Plastic and Reconstructive Surgery, Stoke Mandeville Hospital, Aylesbury, UK
[4]Plastic and Reconstructive Surgery, Wexham Park Hospital, Slough, UK
[5]Neonatal Medicine, School of Public Health, Imperial College London, London, UK
[6]Plastic and Reconstructive Surgery, Imperial College Healthcare NHS Trust, London, UK

**Correspondence to**
Chris Gale;
christopher.gale@imperial.ac.uk

## ABSTRACT

**Objectives**  This systematic review aims to assess the quality of literature supporting surgical interventions for paediatric extravasation injury and to determine whether there is sufficient evidence to support invasive techniques in children.

**Methods**  We performed a systematic review by searching Ovid MEDLINE and EMBASE as well as AMED, CINAHL, Cochrane Central Register of Controlled Trials, Cochrane Database of Systematic Reviews and clinicaltrials.gov from inception to February 2019. Studies other than case reports were eligible for inclusion if the population was younger than 18 years old, if there was a surgical intervention aimed at treating extravasation injury and if they reported on outcomes. Study quality was graded according to the National Institutes of Health study quality assessment tools.

**Results**  26 studies involving 728 children were included—one before-and-after study and 25 case series. Extravasation injuries were mainly confined to skin and subcutaneous tissues but severe complications were also encountered, including amputation (one toe and one below elbow). Of the surgical treatments described, the technique of multiple puncture wounds and instillation of saline and/or hyaluronidase was the most commonly used. However, there were no studies in which its effectiveness was tested against another treatment or a control and details of functional and aesthetic outcomes were generally lacking.

**Conclusion**  Surgical management is commonly reported in the literature in cases where there is significant soft tissue injury but as there are no comparative studies, it is unclear whether this is optimal. Further observational and experimental research evaluating extravasation injuries, including a centralised extravasation register using a universal grading scheme and core outcome set with adequate follow-up, are required to provide evidence to guide clinician decision-making.

## INTRODUCTION

Extravasation is the inadvertent leakage of a vesicant solution from its intended vascular pathway, commonly a peripheral or central vein, into the surrounding tissue with the potential to harm a patient. A vesicant is any medicine or fluid with the potential to cause blisters, severe tissue injury or necrosis if there is leakage.[1] If left untreated, extravasations

### Strengths and limitations of this study

► A systematic review was performed in accordance with the Preferred Reporting Items for Systematic Reviews and Meta-Analyses statement and using methodology outlined in the Cochrane Handbook for Systematic Reviews of Interventions. It was registered on PROSPERO.
► Two authors used a bespoke inclusion/exclusion form to independently assess study eligibility.
► Studies were eligible for inclusion if the population was younger than 18 years old, if there was a surgical intervention aimed at treating extravasation injury in any setting and if they reported on short-term or long-term outcomes.
► Two researchers also independently assessed the included studies' risk of methodological bias using the National Institutes of Health study quality assessment tools.
► 18 years old may represent a relatively arbitrary cut-off age to differentiate between 'paediatric' and 'adult' in terms of extravasation injury.

may result in delayed healing, scarring and functional morbidity, including amputation in severe cases.

Children requiring intravenous (IV) therapy often have multiple risk factors for extravasation injuries, and neonates are at even greater risk of more serious injury due to poor venous integrity, capillary leakage and expandable subcutaneous tissue that accommodates relatively large volumes of fluid.[2] The reported incidence in adult and paediatric inpatients ranges between 0.1% and 6.5% across studies from the UK, USA and Canada, but inconsistent reporting and documentation means that the true figure is likely to be higher.[2–4]

Prevention of extravasation is the foremost priority. However, when injury does occur, multiple treatments have been described. The 'Gault technique'[5] utilises multiple small incisions made around the injury site. These are subsequently used to instil saline and/or

hyaluronidase to wash out the extravasation fluid. This method remains a commonly described technique for the treatment of more severe paediatric extravasations. However, there is a lack of high-quality comparative studies in which treatments are evaluated against each other and/or controls. It is therefore not known which of these treatments are effective, harmful or at the very least, necessary; a conclusion supported by a recent scoping review of interventions.[6] That review highlighted the fact that the lack of consensus extended to published guidelines. Their survey also showed wide variation in the frequency of use of the saline washout techniques in neonatal units across the National Health Service in the UK. This systematic review aims to evaluate the quality of the literature and to summarise the published outcome data of the surgical management of paediatric extravasation injuries.

## METHODS

Our aim was to assess the quality of literature supporting surgical interventions for paediatric extravasation injury and to determine whether there is sufficient evidence to support invasive techniques in children. We performed a systematic review of original clinical studies or systematic reviews (excluding case reports) of paediatric extravasation injury, reported in accordance with the Preferred Reporting Items for Systematic Reviews and Meta-Analyses (PRISMA)[7] statement and using methodology outlined in the Cochrane Handbook for Systematic Reviews of Interventions[8] where applicable. The protocol was developed prospectively, locally peer-reviewed and registered on the PROSPERO database (CRD42016045647). Variability in baseline data, intervention data and outcome data precluded formal meta-analysis. Results are therefore presented as a narrative comparison of different invasive techniques with summary statistics.

### Search methods

Studies were identified through a systematic literature search in Ovid MEDLINE and Ovid EMBASE as well as AMED, CINAHL, Cochrane Central Register of Controlled Trials, Cochrane Database of Systematic Reviews and clinicaltrials.gov from inception to February 2019. To identify grey literature, Web of Science was searched using bespoke searches. Internet-wide searches using Google were employed to identify clinical guidelines describing the management of extravasation in the UK. All of these strategies comprised modifications of a common set of terms to identify the relevant clinical area. Key search terms in this common set included the Medical Subject Headings (MeSH) 'extravasation of diagnostic and therapeutic materials', 'infant', 'child' and 'preschool'. These MeSH terms were combined with appropriate Boolean operators (online supplementary file 1). The index search terms were then combined with free text terms to identify appropriate clinical studies. No language restrictions were imposed. The search results were merged and, after screening, duplicate citations were discarded along with studies unrelated to the research objective.

### Criteria for selecting studies

Study selection criteria were determined in advance during the protocol stage. Two authors used a bespoke inclusion/exclusion form to independently assess eligibility of the title and abstracts of the search results (Online supplementary file 2). Randomised control trials, quasi-experimental and cross-sectional studies and case-series were eligible for inclusion if the population was younger than 18 years old, if there was a surgical intervention aimed at treating extravasation injury in any setting and if they reported on short-term or long-term outcomes. Hyaluronidase injections were considered to be a surgical intervention for the purposes of this review first, because they are an invasive procedure and second, to allow clinically useful comparison to the combination of Gault's technique with hyaluronidase. Any disparities regarding inclusion of articles were discussed between the authors and a joint decision was made based on the inclusion criteria.

### Data analysis

Data extraction was performed by two authors independently and is displayed in tables 1–5. Descriptive analysis for each group of studies was performed, including the number of children with skin necrosis at presentation as well as the difference in outcomes between those who received an intervention and those who received no intervention.

### Study quality assessment

Two researchers also independently assessed the quality of the included studies using the National Institutes of Health study quality assessment tools[9] for 'case series' and for 'before-after studies with no control group'—the two types of included studies. Discrepancies were resolved by discussions with the other authors.

In our review, for a case series to achieve a rating of 'good', it must include a clearly defined study population with comparable subjects as well as clearly described interventions, outcome measures and results with adequate follow-up (questions 2, 4, 5, 6, 7 and 9 in table 6). If the study lacked one of these criteria, it was rated as 'fair' but if it lacked two or more, it was deemed 'poor' and at high risk of methodological bias. A similar scoring system was used for the one before-and-after study, incorporating the different questions in that tool.

### Patient and public involvement

No patients involved.

## RESULTS
### Search strategy

Our search strategy identified a total of 1966 research articles of which 220 were potentially relevant to the research question. Of the 220 papers, 185 did not meet the

**Table 1** Included studies

| Ref | Author | Year | Country | Title | N | Methods | Period | Mean follow-up |
|---|---|---|---|---|---|---|---|---|
| 4 | Murphy et al[4] | 2017 | Australia | Extravasation injury in a paediatric population | 43 | Retrospective series | – | – |
| 10 | Falcone et al[10] | 1989 | USA | Nonoperative management of full-thickness intravenous extravasation injuries in premature neonates using enzymatic debridement | 15 | Retrospective series | – | 6 months |
| 11 | Ghanem et al[11] | 2015 | UK | Childhood extravasation injuries: improved outcome following the introduction of hospital-wide guidelines | 48 | Prospective series | 1 year | – |
| 12 | Kostogloudis et al[12] | 2015 | Greece | Severe extravasation injuries in neonates: a report of 34 cases | 34 | Retrospective series | 2 years | 15 months |
| 13 | Sung and Lee[13] | 2016 | Korea | Nonoperative management of extravasation injuries associated with neonatal parenteral nutrition using multiple punctures and a hydrocolloid dressing | 12 | Retrospective series | 4 years | 10 months |
| 14 | Odom et al[14] | 2018 | USA | Peripheral infiltration and extravasation injury methodology: a retrospective study | 147 | Retrospective series | – | – |
| 15 | Compaña et al[15] | 2017 | Spain | Lesions associated with calcium gluconate extravasation | 4 | Retrospective series | – | 2 months |
| 16 | Harris et al[16] | 2001 | UK | Limiting the damage of iatrogenic extravasation injury in neonates | 56 | Prospective series | 3 years | – |
| 17 | Andrés et al[17] | 2006 | Spain | Treatment protocol for extravasation lesions | 15 | Retrospective series | 6 years | – |
| 18 19 | Linder et al[18 19] | 1983, 1985 | USA | Management of extensive doxorubicin hydrochloride extravasation injuries Prevention of extravasation injuries secondary to doxorubicin | 18 | Retrospective series | – | 3 months |
| 20 | Upton et al[20] | 1979 | USA | Major intravenous extravasation injuries | 7 | Retrospective series | 10 years | – |
| 21 | Casanova et al[21] | 2001 | France | Emergency treatment of accidental infusion leakage in the newborn: report of 14 cases | 14 | Retrospective series | – | – |
| 22 | von Heimburg and Pallua[22] | 1998 | Germany | Early and late treatment of iatrogenic injection damage | 5 | Retrospective series | – | – |
| 23 | Weiss et al[23] | 1975 | Israel | Localized necrosis of scalp in neonates due to calcium gluconate infusions: a cautionary note | 4 | Retrospective series | – | – |

Continued

**Table 1** Continued

| Ref | Author | Year | Country | Title | N | Methods | Period | Mean follow-up |
|---|---|---|---|---|---|---|---|---|
| 24 | Hanrahan[24] | 2013 | USA | Hyaluronidase for treatment of intravenous extravasations: implementation of an evidence-based guideline in a pediatric population | 113 | Before-and-after study | 4 years | – |
| 25 | Boyar and Galiczewski[25] | 2018 | USA | Efficacy of dehydrated human amniotic membrane allograft for the treatment of severe extravasation injuries in preterm neonates | 4 | Retrospective series | – | 1–2 months |
| 26 | Yan et al[26] | 2017 | China | Incidence, risk factors and treatment outcomes of drug extravasation in pediatric patients in China | 18 | Retrospective series | 6 months | 1–5 months |
| 27 | Myers et al[27] | 2018 | USA | Managing intravenous infiltration injuries in the neonatal intensive care unit | 28 | Retrospective series | 7 years | – |
| 28 | Cochran et al[28] | 2002 | USA | Treatment of iodinated contrast material extravasation with hyaluronidase | 8 | Retrospective series | 7 years | – |
| 29 | Cho et al[29] | 2007 | Korea | Successful combined treatment with total parenteral nutrition fluid extravasation injuries in preterm infants | 5 | Retrospective series | 4 months | – |
| 30 | Onesti et al[30] | 2012 | Italy | The use of hyalomatrix PA in the treatment of extravasation affecting premature neonates | 26 | Retrospective series | 6 years | 14 months |
| 31 | Firat et al[31] | 2013 | Turkey | Management of extravasation injuries: a retrospective study | 13 | Retrospective series | 2 years | – |
| 32 | Ching et al[32] | 2014 | UK | Paediatric extravasation injuries: a review of 69 consecutive patients | 69 | Retrospective series | 1 year | 3 days |
| 33 | Sivrioglu and Irkoren[33] | 2014 | Turkey | Versajet hydrosurgery system in the debridement of skin necrosis after calcium gluconate extravasation: report of 9 infantile cases | 9 | Cohort study | – | 1 year |
| 34 | Yan et al[34] | 2014 | China | Treatment of cutaneous injuries of neonates induced by drug extravasation with hyaluronidase and hirudoid | 13 | Retrospective series | 2 years | 3 months |

inclusion criteria. Thirty-five papers were deemed eligible for inclusion after studies with conservative management or no interventions were excluded. The requisite data were immediately available in 26 study reports[4 10–34] and the authors of the remaining nine studies[5 35–42] were contacted twice with additional requests, mainly for exclusively paediatric extravasation data, and given at least 2 months to reply. Unfortunately, none of the authors were able to provide these data and those studies were excluded. The data were extracted from the remaining 26 studies (online supplementary PRISMA flow-diagram) using a prespecified (review-specific) proforma (online

supplementary file 3) by two researchers, with discrepancies resolved by a third.

### Study characteristics
The included studies were published between 1975 and 2018. The mean sample size was 32 with a range of 4–147 participants. One study was a before-and-after comparative study,[24] two were prospective case series[11 16] and the remaining 23 were retrospective case series. The follow-up periods were not reported in 14 included studies. In the other studies, the follow-up periods ranged from 3 days to 15 months (table 1).

 Little M, et al. BMJ Open 2020;**10**:e034950. doi:10.1136/bmjopen-2019-034950

**Table 2** Quality assessment for included studies

**Case series**

| | | Questions | | | | | | | | | Overall rating | Level of evidence |
|---|---|---|---|---|---|---|---|---|---|---|---|---|
| Ref | Study | 1 | 2 | 3 | 4 | 5 | 6 | 7 | 8 | 9 | | |
| 4 | Murphy et al[4] | Y | Y | N | Y | Y | Y | Y | – | Y | Good | 4 |
| 12 | Kostogloudis et al[12] | Y | Y | N | Y | Y | Y | Y | – | Y | Good | 4 |
| 18 | Linder et al[18] | N | Y | Y | Y | Y | Y | Y | – | Y | Good | 4 |
| 19 | Linder et al[19] | N | Y | Y | Y | Y | Y | Y | – | Y | Good | 4 |
| 25 | Boyar and Galiczewski[25] | Y | Y | N | Y | Y | Y | Y | – | Y | Good | 4 |
| 27 | Myers et al[17] | Y | Y | Y | Y | Y | Y | Y | – | Y | Good | 4 |
| 30 | Onesti et al[30] | N | Y | N | Y | Y | Y | Y | – | Y | Good | 4 |
| 34 | Yan et al[34] | Y | Y | N | Y | Y | Y | Y | – | Y | Good | 4 |
| 10 | Falcone et al[10] | Y | Y | N | Y | Y | N | Y | – | Y | Fair | 4 |
| 13 | Sung and Lee[13] | Y | Y | N | Y | Y | N | Y | – | Y | Fair | 4 |
| 23 | Weiss et al[23] | N | Y | Y | Y | Y | N | Y | – | Y | Fair | 4 |
| 29 | Cho et al[29] | Y | Y | N | Y | Y | N | Y | – | Y | Fair | 4 |
| 31 | Firat et al[31] | Y | Y | N | Y | Y | N | Y | – | Y | Fair | 4 |
| 33 | Sivrioglu and Irkoren[33] | Y | Y | N | Y | Y | N | Y | – | Y | Fair | 4 |
| 11 | Ghanem et al[11] | Y | Y | N | Y | N | N | Y | – | Y | Poor | 4 |
| 14 | Odom et al[14] | Y | Y | N | Y | N | Y | N | – | Y | Poor | 4 |
| 15 | Compaña et al[15] | N | N | N | Y | N | N | – | – | Y | Poor | 4 |
| 16 | Harris et al[16] | N | N | Y | – | Y | N | – | – | Y | Poor | 4 |
| 17 | Andrés et al[17] | Y | N | Y | N | Y | Y | – | – | Y | Poor | 4 |
| 20 | Upton et al[20] | Y | N | Y | Y | Y | N | Y | – | Y | Poor | 4 |
| 21 | Casanova et al[21] | Y | Y | N | Y | Y | N | N | – | Y | Poor | 4 |
| 22 | von Heimburg and Pallua[22] | N | N | Y | N | Y | N | Y | – | Y | Poor | 4 |
| 26 | Yan et al[26] | Y | N | Y | – | Y | N | Y | Y | Y | Poor | 4 |
| 28 | Cochran et al[28] | Y | N | N | Y | Y | N | N | – | N | Poor | 4 |
| 32 | Ching et al[32] | Y | Y | Y | Y | Y | N | N | – | Y | Poor | 4 |
| 5 | Gault[5] | N | N | Y | N | Y | Y | N | – | Y | Poor | 4 |

**Before-and-after study**

| Ref | Study | 1 | 2 | 3 | 4 | 5 | 6 | 7 | 8 | 9 | 10 | 11 | 12 | Overall rating | Level of evidence |
|---|---|---|---|---|---|---|---|---|---|---|---|---|---|---|---|
| 24 | Hanrahan[24] | Y | N | – | – | Y | Y | Y | N | – | Y | N | N | Fair | 3 |
| Key | Y | | N | | – | | | | | | | | | | |
| | Yes | | No | | N/A, Not recorded or cannot determine | | | | | | | | | | |

**Case series questions**
(1) Was the study question or objective clearly stated?
(2) Was the study population clearly and fully described, including a case definition?
(3) Were the cases consecutive?
(4) Were the subjects comparable?
(5) Was the intervention clearly described?
(6) Were the outcome measures clearly defined, valid, reliable and implemented consistently across all study participants?
(7) Was the length of follow-upadequate?
(8) Were the statistical methods well described?
(9) Were the results well described?
**Before-and-after study questions**
(1) Was the study question or objective clearly stated?
(2) Were eligibility/selection criteria for the study population prespecified and clearly described?
(3) Were the participants in the study representative of those who would be eligible for the test/service/intervention in the general or clinical population of interest?
(4) Were all eligible participants who met the prespecified entry criteria enrolled?
(5) Was the sample size sufficiently large to provide confidence in the findings?
(6) Was the test/service/intervention clearly described and delivered consistently across the study population?
(7) Were the outcome measures prespecified, clearly defined, valid, reliable and assessed consistently across all study participants?
(8) Were the people assessing the outcomes blinded to the participants' exposures/interventions?
(9) Was the loss to follow-up after baseline 20% or less? Were those lost to follow-up accounted for in the analysis?
(10) Did the statistical methods examine changes in outcome measures from before to after the intervention? Were statistical tests done that provided p-values for the pre-to-post changes?
(11) Were outcome measures of interest taken multiple times before the intervention and multiple times after the intervention (ie, did they use an interrupted time-series design)?
(12) If the intervention was conducted at a group level (eg, a whole hospital, a community) did the statistical analysis take into account the use of individual-level data to determine effects at the group level?

**Table 3** Demographics

| Ref | Author | N | Mean age (months) | Male | Female | N Upper limb | N Lower limb | N Scalp | N other | Peripheral cannula | Central cannula |
|---|---|---|---|---|---|---|---|---|---|---|---|
| **Gault technique studies** | | | | | | | | | | | |
| 4 | Murphy et al[4] | 43 | – | – | – | 32 | 9 | – | – | – | – |
| 11 | Ghanem et al[11] | 48 | 38.4 | – | – | 48 | 25 | 4 | 5 | 73 | 7 |
| 12 | Kostgloudis et al[12] | 34 | 0.6 | – | – | 6 | 28 | – | – | 34 | – |
| 17 | Andrés et al[17] | 15 | 36 | – | – | 14 | – | 1 | – | 15 | – |
| 21 | Casanova et al[21] | 14 | 1 | – | – | 4 | 9 | 1 | – | – | – |
| 32 | Ching et al[32] | 69 | 0.7 | 32 | 37 | 45 | 17 | – | 7 | – | – |
| 16 | Harris et al[16] | 56 | – | – | – | – | – | – | – | – | – |
| | **Total** | **279** | **15.3 (mean)** | **32** | **37** | **149** | **88** | **6** | **12** | **122** | **7** |
| **Debridement+further surgery** | | | | | | | | | | | |
| 10 | Falcone et al[10] | 15 | 0.6 | 7 | 8 | 12 | 2 | 1 | – | – | – |
| 13 | Sung and Lee[13] | 12 | – | 6 | 6 | 7 | 5 | – | – | – | – |
| 15 | Compaña et al[15] | 4 | 2.6 | 3 | 1 | 2 | 2 | – | – | 4 | – |
| 17 | Andrés et al (2)[17] | 15 | 36 | – | – | 14 | – | 1 | – | 15 | – |
| 20 | Upton et al[20] | 7 | 67.2 | – | – | – | – | – | – | – | – |
| 23 | Weiss et al[23] | 4 | – | – | – | – | – | 4 | – | 4 | – |
| 25 | Boyar and Galiczewski[25] | 4 | 1.1 | 3 | 1 | 3 | 1 | – | – | 4 | – |
| 29 | Cho et al[29] | 5 | 0.6 | – | – | 4 | – | – | – | 6 | – |
| 30 | Onesti et al[30] | 26 | 0.6 | 17 | 9 | 14 | 10 | 2 | – | – | – |
| 31 | Firat et al[31] | 13 | 50 | 3 | 10 | 6 | 4 | 3 | – | – | – |
| 33 | Sivrioglu and Irkoren[33] | 9 | 0.9 | – | – | 5 | 3 | 1 | – | 9 | – |
| 18 19 | Linder et al[18 19] | 18 | – | – | – | – | – | – | – | – | – |
| 22 | von Heimburg and Pallua[22] | 5 | – | – | – | – | – | – | – | – | – |
| | **Total** | **137** | **17.7 (mean)** | **46** | **28** | **67** | **27** | **12** | **0** | **42** | **0** |
| **Hyaluronidase injections** | | | | | | | | | | | |
| 14 | Odom et al[14] | 147 | – | 87 | 60 | 106 | 40 | 2 | – | 147 | – |
| 26 | Yan et al[26] | 18 | 39.7 | 10 | 8 | 12 | 6 | – | – | 18 | – |
| 27 | Myers et al[27] | 28 | 1.3 | – | – | 14 | 13 | 1 | – | – | – |
| 28 | Cochran et al[28] | 8 | – | 4 | 4 | – | – | – | – | – | – |
| 34 | Yan et al[34] | 13 | 0.9 | 8 | 5 | 9 | 3 | 1 | – | – | – |
| 24 | Hanrahan[24] | 113 | – | – | – | – | – | – | – | – | – |
| | **Total** | **327** | **14.0 (mean)** | **109** | **77** | **141** | **62** | **4** | **0** | **165** | **0** |
| | **Overall total** | **728** | **15.13 (mean)** | **187** | **142** | **343** | **177** | **21** | **12** | **314** | **7** |

## Study quality assessment results

The one before-and-after study[24] was given an overall 'fair' rating by both authors as it had clearly described objectives, interventions and outcome measures with a large sample size but lacked details on participant eligibility and individual level data. Based on the criteria set out above in the study quality assessment, 11 of the 25 included case series—and the Gault study itself—were rated 'poor', six were 'fair' and eight 'good' (table 2).

## Patient characteristics

Across the 26 included studies, there were 728 children. The median age of the 68 children for whom the individual data were available was 0.66 months (IQR=5.05). The mean age of 14 months (range: neonatal to 17 years old) was calculated using the data for 322 children across 16 included studies. There was a small difference in sex, with 187 males (57%) and 142 females (43%) (data on sex was lacking for 399 children). Of the 26 included studies, 19 recorded the sites of the implicated cannula for 553 children. The most common site was the upper limb (n=343; 62%), followed by the lower limb (n=177; 32%) and then the scalp (n=21; 4%). Ten studies specifically stated whether the cannula was a peripheral (n=314; 98%) or central line (n=7; 2%) (table 3).

Of the 16 studies that provided data on comorbidities for 167 children, prematurity was the most common

**Table 4** Comorbidities and extravasated materials in those papers that recorded them

| Ref | Author | Comorbidities | | | Vesicant type | | | | | | | | Other |
|---|---|---|---|---|---|---|---|---|---|---|---|---|---|
| | | Prematurity | Sepsis | Malignancy | TPN | ≥5% Dextrose | Other IV fluids | Calcium-containing | Antimicrobials | Chemo | Contrast | Dopamine | |
| **Gault technique studies** | | | | | | | | | | | | | |
| 4 | Murphy et al[4] | – | – | – | 10 | 11 | 6 | 2 | 6 | – | 2 | – | 5 |
| 11 | Ghanem et al[11] | 14 | – | – | 22 | 10 | 6 | – | 8 | – | – | 1 | 1 |
| 12 | Kostogloudis et al[12] | 34 | – | – | 28 | 4 | – | – | 2 | – | – | – | – |
| 17 | Andrés et al[17] | – | 1 | 5 | 7 | – | – | 4 | – | 4 | – | – | – |
| 21 | Casanova et al[21] | – | – | – | – | – | – | 2 | – | – | – | 9 | 3 |
| 32 | Ching et al[32] | 'Majority' | – | – | 16 | – | 22 | – | – | – | – | – | 31 |
| | **Total** | **48** | **1** | **5** | **83** | **25** | **34** | **8** | **16** | **4** | **2** | **10** | **40** |
| **Debridement+further surgery** | | | | | | | | | | | | | |
| 10 | Falcone et al[10] | 15 | – | – | 9 | 1 | 3 | 1 | – | – | – | – | 2 |
| 13 | Sung and Lee[13] | 9 | – | – | 12 | – | – | – | – | – | – | – | – |
| 15 | Compaña et al[15] | 2 | – | – | – | – | – | 4 | – | – | – | – | – |
| 17 | Andrés et al[17] (2) | – | 1 | 5 | 7 | – | – | 4 | – | 4 | – | – | – |
| 18 19 | Linder et al[18 19] | – | – | 18 | – | – | – | – | – | 18 | – | – | – |
| 20 | Upton et al[20] | – | – | 1 | – | 1 | 2 | 1 | 1 | 2 | – | – | – |
| 23 | Weiss et al[23] | 4 | – | – | – | – | – | 4 | – | – | – | – | – |
| 25 | Boyar and Galiczewski[25] | 4 | – | – | 4 | – | – | – | – | – | – | – | – |
| 29 | Cho et al[29] | 5 | – | – | 4 | – | – | – | – | – | – | – | 1 |
| 30 | Onesti et al[30] | 26 | – | – | – | – | 26 | – | – | – | – | – | – |
| 31 | Firat et al[31] | – | 1 | 1 | – | 1 | 4 | 3 | 3 | 1 | – | – | 1 |
| 33 | Sivrioglu and Irkoren[33] | 3 | – | – | – | – | – | 9 | – | – | – | – | – |
| | **Total** | **68** | **2** | **25** | **36** | **3** | **35** | **26** | **4** | **25** | **0** | **0** | **4** |
| **Hyaluronidase injections** | | | | | | | | | | | | | |
| 26 | Yan et al[26] | – | – | – | 1 | – | – | 1 | – | – | – | – | – |
| 28 | Cochran et al[28] | – | – | – | – | – | – | – | – | – | 8 | – | – |
| 34 | Yan et al[34] | 5 | – | – | 9 | – | 1 | 1 | – | – | – | – | 2 |
| | **Total** | **5** | **0** | **0** | **10** | **0** | **1** | **2** | **0** | **0** | **8** | **0** | **2** |
| | **Overall total** | **121** | **2** | **25** | **122** | **28** | **70** | **32** | **20** | **25** | **10** | **10** | **46** |

TPN, total parenteral nutrition.

**Table 5** Interventions

| Ref | Author | Intervention summary | Outcomes |
|---|---|---|---|
| **Gault technique studies** | | | |
| 4 | Murphy et al[4] | Gault's+saline (11% of children) | Three children suffered injuries, which led to significant tissue necrosis, delayed healing and prolonged morbidity. None of these were washed out due to delayed referral |
| 12 | Kostogloudis et al[12] | Gault's+saline (100% of children) | Seven children developed superficial blistering and epidermolysis, while six developed necrosis, all post-treatment. All wounds healed within 25 days. One case of distal foot ischaemia resolved after treatment |
| 16 | Harris et al[16] | Gault's+saline (100% of children) | No episodes of skin or soft tissue loss were recorded and no reconstructive surgery was required |
| 17 | Andrés et al[17] (early) | Gault's+saline (67% of children) | Seven of 10 treated with Gault's technique avoided necrosis and recovered fully. Three developed minor necrosis. Tthe remaining five were debrided and received artificial skin and obtained satisfactory outcomes |
| 21 | Casanova et al[21] | Gault's+hyaluronidase (79% of children)/+saline (14%) with liposuction | No skin involvement in 10 children; blistering healed in one; necrosis resolved in three |
| 11 | Ghanem et al[11] | Gault's+hyaluronidase (46% of children) with liposuction | Three children had tissue necrosis—two were late referrals; unclear if the other one received washout. There was satisfactory healing with no requirement for surgical intervention |
| 32 | Ching et al[32] | Gault's+hyaluronidase (62% of children) | Of the 62% of children washed out, none developed complications. One calcinosis cutis and one ischaemic toe requiring amputation among children receiving no treatment |
| **Debridement+further surgery** | | | |
| 10 | Falcone et al[10] | Topical fibrinolysin/deoxyribonuclease ointment then debridement | All wounds healed completely with no infections and no functional scar contractions at up to 16 months follow-up. No skin grafts were required |
| 23 | Weiss et al[23] | Wet dressings and repeated economical debridement | Wounds healed well in 15–40 days. Scars were visible but without discolouration |
| 31 | Firat et al[31] | Topical hirudin and antibiotics, then 3% boric acid, then repetitive debridement | Seven children required split-thickness skin grafting and two required fasciocutaneous flaps. All recovered well, with scar development in four. Minor functional loss in the hands or feet as a result of scar formation was managed by physiotherapy and pressure garments |
| 30 | Onesti et al[30] | Topical collagenase, then debridement and then hyalomatrix PA (dermal substitute) | 18 children healed fully after 21 days. Four had pathological scars and four had debilitating scar contractures needing secondary surgery |
| 25 | Boyar and Galiczewski[25] | Enzymatic or autolytic debridement before mechanical debridement and application of dehydrated human amniotic membrane allograft (dHAMA) | Complete closure of significant wounds with minimal soft scars and normal pigmentation |
| 15 | Compaña et al[15] | Topical steroids, Burow's solution and silver sulfadiazine for all children. 3 (60%) then underwent debridement followed by split-thickness skin grafts | Successful healing in two children. One died of other causes |

Continued

**Table 5**  Continued

| Ref | Author | Intervention summary | Outcomes |
|---|---|---|---|
| 29 | Cho et al[29] | Topical antibiotic+anti-inflammatory herbal mixture for all children. 1 (20%) debridement. 1 (20%) escharotomy | The child who underwent debridement had a small-sized contracture at 50 days |
| 20 | Upton et al[20] | Debridement and skin grafts. Excision of extensor tendons if infected or devascularised. All children required two or more operations | Two children experienced contractures, two had extensor loss, one had hair loss, one had loss of motion and one required further reconstruction |
| 18 19 | Linder et al[18 19] | Debridement and wound closure: mostly split-thickness skin grafts or delayed primary closure. All children had at least two operations | The mean time for wound closure was 49 days (range 10–85 days). Three children died before wound closure. At least one patient needed a split-thickness skin graft. One child developed sympathetic dystrophy syndrome. Some children developed permanent joint stiffness |
| 22 | von Heimburg and Pallua[22] | Debridement, allogeneic donor tissue grafts and autologous split-thickness skin grafts | After 15 days there was full healing in all five infants |
| 17 | Andrés et al[17] (late) | Debridement+dermal substitute in 33% | The five late referrals were debrided and received artificial skin. All obtained satisfactory outcomes |
| 13 | Sung and Lee[13] | Multiple punctures using a scalpel blade+hydrocolloid dressing. Then debridement | All children showed favourable results without functional deficits or conspicuous scars |
| 33 | Sivrioğlu and Irkoren[33] | Versajet hydrosurgery for all children. 1 (11%) sharp debridement | Minimal scar formation with no hypertrophic scarring in any patient |
| **Hyaluronidase injections** | | | |
| 14 | Odom et al[14] | Injection of hyaluronidase or phentolamine without incisions | No children required surgical intervention for wound healing or had an infection |
| 24 | Hanrahan[24] | Injection of hyaluronidase | Mean harm scores were similar between the group receiving hyaluronidase and the group not receiving it |
| 26 | Yan et al[26] | Injection of hyaluronidase (33%). 1 (6%) required surgical excision of a lesion | All healed and had 'good outcomes' |
| 27 | Myers et al[27] | Injection of hyaluronidase in 50% | Time to healing averaged 16.2 days (range 1–82 days). No patient required surgical intervention |
| 28 | Cochran et al[28] | Injection of hyaluronidase in 25% | One patient had a prolonged course with swelling and skin peeling of the hand |
| 34 | Yan et al[34] | Application of hirudoid and injection of hyaluronidase | Three children lost to follow-up. Negligible loss of functional movements. One case of scarring and readmission with calcinosis |

comorbidity in 11 (n=121; 72%). Malignancy (n=25; 15%) was the main comorbidity in three studies.

Of the 20 studies that reported on them, the most commonly extravasated substances affecting 363 children were (table 4) as follows:

1. Total parenteral nutrition (TPN)—122 children (34%); most common in eight studies.
2. IV maintenance fluids other than those specifically listed—70 children (19%); most common in three studies. Unfortunately, the data were not available for the outcomes of different types of maintenance fluids so they have been grouped together.
3. Calcium-containing products—32 children (9%); most common in four studies.
4. Dextrose of concentration ≥5%–28 children (8%); most common in one study.
5. IV antimicrobials—20 children (6%); most common in no studies. Of those 20 extravasations, five were due to flucloxacillin—with one injury taking 60 days—to heal and five were due to unspecified cephalosporin antibiotics, three of which required skin grafting. One case of severe contractures and extensor loss on the dorsum of a child's hand was due to an unspecified tetracycline antibiotic. There were no individual outcomes recorded for the seven extravasations of aciclovir, the single extravasations of gentamicin, ganciclovir or the remaining extravasations involving flucloxacillin and cephalosporin antibiotics.

**Table 6** Injuries to children receiving an intervention in those papers that recorded them

| | | Skin changes | | | Other injuries | | | | |
|---|---|---|---|---|---|---|---|---|---|
| Ref | Author | Pain/ swelling/ erythema | Partial thickness skin injury | Full thickness skin injury | Tendon | Nerve | Vascular | Compartment Syndrome | Amputation |
| **Gault technique studies** | | | | | | | | | |
| 12 | Kostogloudis et al[12] | – | 7 (post) | 6 (post) | – | – | 1 (pre) | 1 (pre) | – |
| 17 | Andrés et al[17] (early) | – | – | 3 (post) | – | – | – | – | – |
| 21 | Casanova et al[21] | 10 (pre) | 1 (pre) | 3 (post) | – | – | – | – | – |
| | **Total** | **10** | **8** | **12** | **0** | **0** | **1** | **1** | **0** |
| **Debridement+further surgery** | | | | | | | | | |
| 10 | Falcone et al[10] | – | – | 16 (pre) | – | – | – | – | – |
| 13 | Sung and Lee[13] | 1 (pre) | 2 (pre) | 2 (pre)+7 (post) | – | – | – | – | – |
| 15 | Compaña et al[15] | – | – | 3 (pre) | – | – | – | – | – |
| 17 | Andrés et al[17] (late) | – | – | 5 (pre) | – | – | – | – | – |
| 18 19 | Linder et al[18 19] | – | – | 18 (pre) | – | – | – | – | – |
| 20 | Upton et al[20] | – | – | 7 (?) | 2 (?) | – | – | – | 1 (?) |
| 22 | von Heimburg and Pallua[22] | – | – | 5 (pre) | – | – | – | – | – |
| 23 | Weiss et al[23] | – | – | 4 (pre) | – | – | – | – | – |
| 25 | Boyar and Galiczewski[25] | – | – | 4 (pre) | – | – | – | – | |
| 29 | Cho et al[29] | – | – | 2 (pre) | – | – | – | – | – |
| 30 | Onesti et al[30] | – | 18 (pre) | 8 (pre) | – | – | – | – | – |
| 31 | Firat et al[31] | – | – | 9 (pre) | – | – | – | – | – |
| 33 | Sivrioglu and Irkoren[33] | – | – | 9 (pre) | – | – | – | – | – |
| | **Total** | **1** | **20** | **100** | **2** | **0** | **0** | **0** | **1** |
| **Hyaluronidase injections** | | | | | | | | | |
| 28 | Cochran et al[28] | – | 1 (post) | – | – | – | – | – | – |
| 26 | Yan et al[26] | 17 (pre) | – | 1 (pre) | – | – | – | | |
| 34 | Yan et al[34] | 5 (pre) | 3 (pre) | 1 (pre) | – | – | – | – | – |
| | **Total** | **22** | **4** | **2** | **0** | **0** | **0** | **0** | **0** |
| | **Overall totals** | **33** | **32** | **114** | **2** | **0** | **1** | **1** | **1** |

'?', unclear whether preintervention or postintervention.
'pre', preintervention.
'post', postintervention.

## Interventions

Uncertainty remains over whether there is a difference in the outcomes of children who are treated conservatively or surgically and the case series that exist on conservative management cover a number of treatments for a range of injuries from swelling to full-thickness defects. However, a recent scoping review found that the outcomes used and results detailed were generally limited,[6] making it difficult for clinicians to rely on those studies in deciding who to treat conservatively. This is why we have focused solely on studies describing invasive interventions: the Gault technique, debridement and further surgery, and hyaluronidase injections. We have still reported on the outcomes of children within each group of studies who received no treatment in order to provide a clinically useful comparison to each intervention.

### The Gault technique

Two hundred and seventy-nine children with an average age of 15.3 months were involved in the seven studies looking at the Gault technique. No patients had skin necrosis at presentation.

One hundred and seven were treated using the Gault technique with saline, of which nine (8%) developed full thickness skin injury/skin necrosis, one (1%) had compartment syndrome at presentation requiring fasciotomies and one (1%) had distal foot ischaemia which resolved after treatment. Of those with necrosis, six required no surgical intervention and three were debrided and treated with artificial dermis.

Seventy-six were treated using the Gault technique with hyaluronidase, of which three (4%) developed skin necrosis and were all managed conservatively.

One child received only hyaluronidase injections and developed necrosis, which healed spontaneously.

Ninety-five children received no treatment, of which 10 (11%) developed necrosis (one of whom went on to lose three toes), three (3%) developed an associated infection, one (1%) had an ischaemic toe requiring amputation and one (1%) developed calcinosis cutis (table 5).

### Debridement and further surgery

There were 137 children with an average age of 17.7 months across the 14 studies in the debridement and further surgery group.

Nine studies (98 children) used management plans comprising debridement and further surgery in combination with conservative measures, details of which are shown in table 4.[10 13 15 17 23 25 29–31] Of those 98 children, 34 (35%) had established full thickness skin injury/skin necrosis at presentation and 28 (29%) developed necrosis after conservative measures. Eighty-four children (86%) underwent mechanical debridement as part of their treatment. The vast majority of children's wounds healed with minimal scarring and no functional deficit but four (4%) had debilitating scar contractures necessitating further surgery, 10 (10%) required skin grafts, and two (2%) required fasciocutaneous flaps.

The other five studies (39 children) used only debridement and further surgery. All of those 39 children had full thickness skin injury/skin necrosis at presentation. Thirty (77%) underwent mechanical debridement, of whom 22 (56%) required further operations including skin grafts and fasciocutaneous flaps and two developed contractures. The remaining nine children (23%), all from one study, underwent Versajet hydrosurgery with good healing in all children and one requiring sharp debridement.

In total, of the 137 children in the debridement and further surgery group, 103 (75%) developed skin necrosis, all of whom (in addition to some of those who were yet to develop necrosis, total n=123 (89%)) received either mechanical or water jet debridement. At least 38 children (28%) went on to require further operations.

### Hyaluronidase injections

The final group of studies used hyaluronidase injections at the extravasation site without the need for incisions or washouts. This group comprised six studies and 327 children.[14 24 26–28 34]

Ninety-seven children received hyaluronidase injections, one (1%) of whom had skin necrosis at presentation that resolved and one (1%) developed necrosis requiring further surgery. Of the remaining 230 children that did not receive hyaluronidase, none developed skin necrosis at any stage, although reporting on the extent of skin damage and eventual outcomes across the studies was generally poor.

## Comparison of the interventions
### The Gault technique

Among the studies included in this review, Ghanem et al[11] provides the closest approximation of a comparative study of the Gault technique in children: 48 extravasations were diagnosed and children were divided into early (<24 hours, n=45) and later (>24 hours, n=3) referrals. Among early referrals, 22 were deemed to be 'at risk' injuries and received washout using Gault's technique. Skin necrosis occurred in one of 45 early referrals but in two of three later referrals. The conclusion of this study, that the incidence of necrosis is higher in the later referral group, is limited by low methodological quality including the use of an arbitrary 24 hours cut-off, the small size of the later referral group and a lack of data describing the types of vesicants. It is important to note that the necrosis resolved with conservative management in all three cases.

Of the 183 children treated with a variation of the Gault technique across the included studies, 12 (7%) developed skin necrosis (none had skin necrosis at presentation). This compares favourably to the 11% (10/95) of children who received no treatment across those same studies. However, 73% (69/95) of the untreated group were late referrals and may be expected to have worse outcomes related to time of presentation. The only study to apply the Gault technique in which plastic surgery review time and patient ages were similar was Ching et al.[32] In that study, none of the 43 children treated with Gault's technique and hyaluronidase developed any complications whereas there were three cases of associated infection, one ischaemic toe requiring amputation and one case of calcinosis cutis among the 26 children that received no treatment.

The degree of injury was only recorded for 36 (13%) of the 279 children in the included studies describing the Gault technique, of which 22 (8%) were full-thickness skin injuries.

### Debridement and further surgery

Seventy-three children (53%) of the 137 in the debridement and further surgery studies had at least skin necrosis/full thickness skin injury at presentation.

In contrast to the Gault as well as the hyaluronidase injection studies, the degree of injury was documented for 93% of the children (128/137).

### Hyaluronidase injections

One (0.3%) of the 327 children reported in studies describing hyaluronidase injections had skin necrosis

at presentation, and one (0.3%) went on to develop it. The degree of injury was recorded for only 49 (15%) of the 327 children. The reporting of patient outcomes was particularly scarce in this group.

### Injuries

The types of injury were recorded for 197 (27%) of the 728 children across the included studies. Injuries were limited to pain, swelling or erythema (n=34; 17%) or partial-thickness skin injury such as superficial blistering in others (n=35; 18%) and full-thickness skin injury such as necrosis in the largest number of children (n=121; 61%). However, these figures are skewed by the fact that 10 studies only looked at extravasation injuries causing at least skin necrosis.[10 12 18–20 22 23 25 31 33] Two studies described toe amputations secondary to ischaemia.[4 32] One study of major injuries with full-thickness tissue loss had several casualties including a below elbow amputation and two episodes of extensor loss.[20]

Table 6 displays only the recorded injuries suffered by children who received an intervention. It also states whether those injuries were preintervention or postintervention or whether the timing was unclear. The overall figures are similar to those above, with injuries recorded for 184 (25%) of the children. Of that 184, 33 (18%) were pain, swelling or erythema, 32 (17%) were partial-thickness and 114 (62%) were full-thickness injuries. One study described a transient episode of foot ischaemia that resolved after washout and one compartment syndrome of the left forearm requiring fasciotomies, with washout through the fasciotomy wounds.[12]

### Millam's grading system for extravasation injuries

Several of the included studies[12 14 21 26 29 33] make reference to the four stages of extravasation/infiltration severity, which was first published by Millam in 1988.[43] This focuses on presence of pain, erythema, oedema, capillary refill, skin temperature and breakdown and it has been proposed that stages III and IV may require intervention.[44] Of those studies, three state that all of the children had stage III or IV injuries before any intervention,[12 14 29] and two made reference to the grading scheme without using it to assess the injuries.[21 33]

In one case series,[26] treatment was determined by the severity of extravasation injury based on a grading system similar to Millam's. Of the 18 children, the seven with grade I and II injuries were managed conservatively. Ten children had grade III injuries—five were managed conservatively but the other five who had also developed skin erythema received active treatment with hyaluronidase injections. The one child with a grade IV injury required excision of the lesion—the sole surgical intervention in that study—after having been injected with hyaluronidase. All wounds healed with 'good outcomes'.

Other than the presence of necrosis signifying a grade IV injury, the other included studies described injuries in ways that did not easily translate into Millam's grading

system. This led to the authors of this study using the injury descriptors seen in the table 5 headings.

### DISCUSSION

This paper describes a systematic review of characteristics and operative management of paediatric extravasation injuries. Our review highlighted three key issues. First, surgical management is commonly reported in the literature in cases where there is significant soft tissue injury but as there are no comparative studies, it is unclear whether this is optimal. Second, there are no data to support one surgical management approach over another, or over no treatment. Third, there are no data on the adverse events caused by the different surgical interventions.

A variation of the Gault technique was used in the largest number of children to receive surgical treatment across the included studies. Gault[4] described a retrospective case series of 96 patients, ranging in age from newborn to 70 years, who were split into two groups: those seen within 24 hours of the extravasation and those seen later than this. Patients seen within 24 hours had treatment as per the Gault technique while those seen after 24 hours had no treatment. The original Gault paper was not included in our review because there were no separate paediatric data available but a separate study quality assessment by two authors rated the study as 'poor' due to the lack of a clearly defined objective or population, subjects who were not comparable and an inadequate length of follow-up (Table 6). Despite the inherent limitations in the original Gault study, the method he described is widely used, as demonstrated by this review where it was a commonly described technique to manage extravasation injuries. This approach is not without downsides: it is an invasive procedure with associated morbidity such as pain and possible tendon and nerve damage, and takes considerable time to perform.

### Evidence for the continuing use of the 'Gault technique'

The degree of injury was recorded for 13% of the children included in the Gault technique studies, 93% of those in the debridement and further surgery group and 15% of those in the hyaluronidase injection group. These figures demonstrate some of the difficulties faced in drawing comparisons between different interventions, both due to the poor recording of injuries and because of the fact that accurate documentation of injury severity appears to be more consistently applied following more severe injuries. To add to this, table 6 demonstrates the interstudy variability on whether injuries were recorded preintervention or postintervention with two studies even reporting a mixture of both separate to the eventual patient outcomes.[13 21] Even when the data are presented in this way, it does not take into account how soon after the intervention the injury has taken place. For example, in several studies, skin necrosis developed 1–2 days after the intervention but then resolved soon afterwards.[12 17 21] This highlights the overall lack of clarity in the reporting of injuries and outcomes. It was therefore not possible to

perform pooled analysis from included studies of Gault and alternative techniques due to study heterogeneity and incomplete outcome reporting.

All but one of the included studies were case series and the one before-and-after study[24] primarily examined the effect of new extravasation treatment guidelines on knowledge of hyaluronidase, extravasation incident reporting and time from extravasation discovery to treatment. Therefore, even though they found similar harm scores for the hyaluronidase and the non-hyaluronidase groups, the results could not be relied on to assess the effectiveness of the intervention.

In addition, inclusion criteria varied considerably, details of the injuries were poorly described and no studies made a comparison to another intervention. Finally, detailed outcome measures following interventions were generally lacking—many studies used subjective descriptions (such as 'healed completely', 'healed fully' or 'no hypertrophic scarring') on which it is difficult to base decisions in clinical practice. In summarising the included studies, it can be noted that despite widespread use, there is no high-quality evidence to support the Gault approach or its modifications.

### The future of extravasation management

A key conclusion from this review is that resolution of extravasation injuries is common regardless of the treatment applied. In the absence of data to support one treatment modality over another, it is imperative that high-quality methodology randomised comparisons are undertaken; however, given the high rate of resolution with conservative treatment, any such trial will require a large, collaborative and multidisciplinary approach.

Although many of the in the included studies required minimal intervention, full-thickness skin injury was commonplace; this has the potential for lifelong scarring in patients with an immature skin barrier such as children and preterm infants in particular. More serious complications were rare, but given that only a minority of studies reported on longer term outcomes, these may underestimate the true burden of extravasation injuries. We suggest the creation of a core outcome set for extravasation injury treatment and that future studies use longer term follow-up to monitor those outcomes.

Creation of a centralised extravasation register would permit larger scale assessment of the available interventions and the use of a universal grading scheme (such as Millam's)[43] to assess the initial extent of injuries would also aid in their comparison. However, this requires meticulous documentation of the appearance of the extravasated site as well as the extent of tissue damage.

This review was limited by the low methodological quality of included studies and inconsistent outcome reporting. Variability in baseline data, intervention data and outcome data precluded formal meta-analysis. These limitations prevent the authors from providing definitive clinical recommendations either for or against paediatric extravasation, based on the current evidence.

## CONCLUSION

There is a paucity of evidence to inform surgical treatment of paediatric and neonatal extravasation injuries. The Gault technique is a commonly described approach, but we did not identify any evaluation of its effectiveness compared with other approaches or conservative management. A centralised extravasation register, use of a standardised and agreed grading scheme, development of a core outcome set and adequate follow-up of extravasation injuries are required to provide incidence and outcome data for this condition. This would inform a much-needed, definitive trial of therapeutic approaches to extravasation injuries in children and neonates.

**Contributors** ML was one of the authors to independently assess eligibility of the title and abstracts of the search results. He then aided in extraction and analysis of the relevant data from included studies, performed the study quality assessments and drafted all of the versions of the article leading up to this submission. SD also aided in the extraction and analysis of the relevant data from included studies, performed the study quality assessments and assisted in the drafting of each version of the article. JCRW made substantial contributions to the design of the study, aided in the extraction and analysis of the relevant data, as well as reviewing and critically editing each version of the article. MG made substantial contributions to the design of the study as well as reviewing and critically editing each version of the article. CG acted as one of the cosenior authors, lending his expertise in paediatrics. He reviewed and critically edited the later versions of the article, substantially changing the text and ensuring its accuracy. AJ acted as the other cosenior author, lending his expertise in plastic surgery. He made substantial contributions to the design of the study, aided in the analysis and interpretation of the data from included studies and critically reviewed each version of the article.

**Funding** JCRW is an NIHR Academic Clinical Fellow; Chris Gale is funded by a Medical Research Council (MRC) Clinician Scientist Fellowship. This research is supported by the National Institute for Health Research (NIHR) infrastructure at NDORMS. The views expressed are those of the author(s) and not necessarily those of the NHS, the NIHR or the Department of Health. No funding was directly received for the development of this manuscript.

**Competing interests** CG has received support from Chiesi Pharmaceuticals to attend an educational conference; in the past 5 years he been investigator on received research grants from Medical Research Council, National Institute of Health Research, Canadian Institute of Health Research, Department of Health in England, Mason Medical Research Foundation, Westminster Medical School Research Trust and Chiesi Pharmaceuticals; he declares no other conflict of interest.

**Patient and public involvement** Patients and/or the public were not involved in the design, or conduct, or reporting, or dissemination plans of this research.

**Patient consent for publication** Not required.

**Provenance and peer review** Not commissioned; externally peer reviewed.

**Data availability statement** All data relevant to the study are included in the article or uploaded as supplementary information. All data are included in the article or uploaded as supplementary information. No additional data are available.

**ORCID iDs**
Max Little http://orcid.org/0000-0001-5627-210X
Justin Conrad Rosen Wormald http://orcid.org/0000-0001-6197-4093
Chris Gale http://orcid.org/0000-0003-0707-876X

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
