## [Reviewer comments · BMJ Open]

ARTICLE DETAILS

TITLE (PROVISIONAL)	Surgical Intervention for Paediatric Infusion-Related Extravasation Injury: A Systematic Review
AUTHORS	Little, Max; Dupré, Sophie; Wormald, Justin; Gardiner, Matthew; Gale, Chris; Jain, Abhilash

VERSION 1 – REVIEW

REVIEWER	Mark Corbett Centre for Reviews and Dissemination, University of York, UK
REVIEW RETURNED	29-Nov-2019

GENERAL COMMENTS	This review was generally well conducted but needs more detail and clarity in specific areas: I think a more detailed description of the eligibility of studies in terms of interventions is needed for clarity - some would not consider a hyaluronidase injection alone as a 'surgical intervention'. Also, on p15 it says "Of the remaining 230 children that did not receive hyaluronidase, none developed skin necrosis..." and on P13 line 37 it says: "95 children received no treatment....." and gives results for these patients. P19 discusses children who were managed conservatively. The review inclusion criteria stated that only surgical interventions were included – should these results be here? (p13 also says the review focus is "solely on surgical interventions"). Please clarify. P19 Risk of bias – I would call this "quality assessment" rather than risk of bias as these are not comparative studies. The nine quality assessment criteria need stating, to allow readers to interpret what Table 6 really means. Similarly, please state how you arrived at the ratings of 'good', 'fair' or 'poor'. I wonder whether some of the quality assessments need revisiting. I was particularly surprised at how many studies got a 'Y' for the "outcomes clearly defined, valid and reliable" (item 6) and "Results well described" (item 9) questions. I would be interested to see what information this was based on (i.e. it would be useful to see justification of decisions, if they are available) so it can be reconciled with the discussion: "poor recording of injuries", "injuries were poorly described", "incomplete outcome reporting" and "detailed outcome measures following interventions were generally lacking" are all mentioned on p22. The limitations of the QA tool should be mentioned as a limitation of the review (though this is not the reviewers' fault, as I don't think there are many QA tools for case series). As the abstract states, functional and aesthetic outcomes were generally lacking in the included studies yet we would expect these important outcomes to
---

	be reported in studies on extravasation injuries in children – was this adequately captured by the QA tool? perhaps comment on things like how applicable the outcomes were to clinical practice, and whether outcome subjectivity is a concern with respect to the 'healed fully', 'healed completely' type of descriptions used by many studies? Minor point: Table 4 Casanova 2001 – wasn't liposuction also used in this study?
--	---

REVIEWER	Samantha Keogh Queensland University of Technology SK's previous and current employer have received monies on her behalf from BD Medical for educational research consultancies, investigator initiated grants and unrestricted grants in aid.
REVIEW RETURNED	27-Jan-2020

GENERAL COMMENTS	Thank you for giving me the opportunity to review this very well conducted review and well written manuscript. Congratulations to the authors for tackling such a comprehensive review of studies since database inception. The strength of the study is the detail with which the authors describe the studies including interventions and outcome measures. The limitations are related to the poor reporting of the included studies, as noted in manuscript. The method is appropriately described, save a need for explicit study aim (rather than a question) - as denoted best in opening of discussion. Also, the authors were remiss in not including description of data analysis in method - however simple. The manuscript therefore would benefit from clarification of study aim and analysis. There are other minor points for possible consideration as indicated in 'sticky note' comments on attached pdf. I recommend publication subject to minor revision.
--

VERSION 1 – AUTHOR RESPONSE

In response to Reviewer 1:

- p6 - we have included our reasoning for the inclusion of hyaluronidase injections
- p13 - we have included our justification for including the data of children receiving 'no treatment' - in order to compare to subjects from the same studies receiving interventions. We have not included studies on conservative management, instead reporting data of children who received no treatment in studies of invasive techniques. We believe that our separate reporting of outcomes for those receiving 'no treatment' also helps to justify this.
- pp19-20 - on review, we agreed with your assessment that too many studies received a 'Y' in certain columns. Although results were generally well-described (item 9), most studies fell at the previous hurdle of having poorly-defined outcomes (item 6). The effect of this in clinical practice is now mentioned on p22. We have also included a more detailed justification of our decisions of 'good', 'fair' and 'poor'.
- 'Risk of bias' has been changed to 'study quality assessment' throughout the paper
- Table 4 amended to specifically mention the studies in which liposuction technique was used

In response to Reviewer 2:

- Abstract - wording of objective and method altered. Completely agreed with the comment re p21 line 9 and have added to the abstract conclusion.
- p4 - perhaps not first line therapy universally but is the first method described in the GOSH

information on extravasation injury. Mentioned first only because that is where we feel the controversy lies and we wanted only to look at invasive interventions.

- p5 - we have added an explicit study aim to our methods

- p6 - we have added a description of our (very simple) data analysis and how the results are presented.

- p15 - unfortunately, detailed data on which infusate caused which injury were generally lacking. Individual cases who fared worse were often the only ones described in detail.

p23 - we have made reference to the registry a little earlier than the conclusion

VERSION 2 – REVIEW

REVIEWER	Mark Corbett University of York, UK
REVIEW RETURNED	13-Mar-2020

GENERAL COMMENTS	P17 I find it difficult to interpret the “Injuries” section and Table 5 – is this section describing the injuries before intervention or after intervention, or both? A clearer description in the text (including the heading) and in the Table 5 heading is needed. Similarly, is the Millam’s grading system section describing injuries before intervention or after intervention, or both? It only gives proper details for one study - where are the injury descriptions pre-surgical intervention for the other studies? Many were doubtless poorly reported (as mentioned in the discussion) but this needs properly documenting (ideally in a table, perhaps Table 2) as it is a key population characteristic. Study quality assessment section p20 - it is good that details have been added explaining how judgments were arrived at, but these should be moved to the Methods section of the manuscript. In a systematic review it is usual for the quality assessment results (Table 6) to come before the main effectiveness results (see PRISMA statement).
---

REVIEWER	Samantha Keogh Queensland University of Technology, Brisbane QLD Australia
	My current and past employer have received on my behalf monies from BD Medical for educational and research consultancies, investigator initiated and unrestricted grants.
REVIEW RETURNED	14-Mar-2020

GENERAL COMMENTS	Thank you for addressing comments and queries from reviewers. This clarification and added detail has enhanced the manuscript and I recommend it for publication.
---

VERSION 2 – AUTHOR RESPONSE

In response to Mr Corbett:

- The study quality assessment has been moved to the Methods section and its results (including what was Table 6, now Table 2) have been moved to the appropriate place before the main effectiveness results.

- Table 5 (now table 6) has been updated with whether the injuries occurred pre- or post-intervention. This was done most clearly by adding this in brackets next to each figure. Reference has been made to this both in the results and in the discussion.
- The table has also been updated to include only the injuries for those that went on to receive an intervention. The results including those who did not receive an intervention are still described in the text for comparison.
- For the six studies that mentioned Millam's grading system, the text has been updated to include whether this was used pre- or post-intervention or whether mere mention was made to the system. Unfortunately it was not widely used and requires detailed documentation, so other descriptors such as 'partial thickness' and 'full thickness' injuries continued to be more useful.

In response to Professor Keogh:

- Thank you for your recommendation for publication